# The Targets for Stunting Prevention Policies in Papua, Indonesia: What Mothers’ Characteristics Matter?

**DOI:** 10.3390/nu14030549

**Published:** 2022-01-27

**Authors:** Ratna Dwi Wulandari, Agung Dwi Laksono, Ina Kusrini, Minsarnawati Tahangnacca

**Affiliations:** 1Faculty of Public Health, Universitas Airlangga, Surabaya 60115, Indonesia; 2The Airlangga Centre for Health Policy (ACeHAP), Universitas Airlangga, Surabaya 60115, Indonesia; agung.dwi.laksono-2016@fkm.unair.ac.id; 3The National Agency for Research and Innovation of the Republic of Indonesia, Jakarta 15412, Indonesia; dyy_syg@yahoo.com; 4Faculty of Health Science, Syarif Hidayatullah Jakarta State Islamic University, Jakarta 15412, Indonesia; minsarnawati@uinjkt.ac.id

**Keywords:** stunted, under five, nutritional status, public health nutrition, public health

## Abstract

The study aimed to analyze the most appropriate maternal characteristics for stunting prevention policies. The study employed secondary data from the 2017 Indonesia Nutritional Status Monitoring Survey. The study obtained weighted samples of 11,887 Papuan children under five years of age. On the other hand, the study used the nutritional status as an outcome variable and maternal characteristics as an exposure variable. The research employed the following four control variables: residence, region, under-five age, and gender. The study occupied the binary logistic regression. The results show that mothers who graduated from primary school and under were 1.263 times more likely than mothers with a college education to have stunted children. Mothers who graduated from junior high school are 1.222 times more likely than mothers with a college education to have stunted children. Mothers who graduated from senior high school were 1.122 times more likely than mothers with a college education to have stunted children. Mothers with a never-married status have a 1.138 times greater probability than divorced/widowed mothers to have stunted children. Meanwhile, married mothers are 0.936 times more likely than divorced/widowed mothers to have stunted children. The study concluded that the target group for stunting prevention policies are mothers with poor education and who are single.

## 1. Introduction

Stunting remains the most common form of malnutrition in children, particularly in poor and developing countries [1]. Stunting is a chronic malnutrition condition manifested by a height of less than −2 standard deviations from the WHO global reference for children compared to others of their age [2]. Stunting is a linear growth disorder characterized by a lack of nutritional quality during the first 1000 days of life [3]. Stunting not only reflects the problem of short stature, but it also reflects malnutrition between generations. If it is not corrected, there will be a steady state of malnutrition from time to time [3,4,5].

Stunting (along with wasting and being underweight) is a problem of malnutrition in children and has a severe impact on the future quality of human resources [6]. Furthermore, malnutrition causes more than 1 million child deaths worldwide, with 3.9% resulting in lives lost and 3.8 percent being the DALY (daily adjusted life years) figure. Unlike wasting and underweight, which are acute nutritional problems, stunting is chronic malnutrition that cannot overcome the effects and causes in the short term [7]. This stunting issue is a serious problem related to the optimal quality of a nation’s human resources [8]. In the long term, stunting in toddlers results in a higher risk of non-communicable diseases due to metabolic disorders in adulthood [9]. The condition will contribute to the burden of diseases. Furthermore, stunted toddlers are more susceptible to infectious diseases [10], interfering with children’s growth and development and adult productivity [8].

Globally, nearly 151 million children are stunted [11]. The figure is considered quite high, so efforts at the global level continue to be carried out. According to the 2025 Global Nutrition target, we expect that the prevalence of stunting in the world will decrease by 20% [12]. The target is still quite difficult to achieve considering the majority of stunting is found in Indonesia, which is still relatively high in the last decade. Almost 3 out of 10 children under five in Indonesia experience stunting, as shown from the National Riskesdas survey data from 2007, 2013, 2018 [13].

Indonesia is currently carrying out stunting prevention efforts through the stunting reduction acceleration team, through which specific and sensitive nutrition intervention efforts continue to be carried out [14,15]. In this case, stunting prevention is not only the health sector’s responsibility but also involves the convergence of cross-sectoral efforts [14,15]. This effort is crucial, considering that the first 1000 days of life is a golden period of child growth, during which an increase in the number and capacity of cells occurs very quickly [16]. Adequate nutrition is critical to achieving optimal growth at this time.

However, this effort cannot be separated from the WHO UNICEF child growth theory framework, where the presence of infectious diseases and the quality of food intake directly influence the nutritional status of children [17]. Therefore, prevention of infectious diseases, immunization, and healthy fulfilment through exclusive breastfeeding and complementary feeding are specific nutrition intervention strategies that can be implemented [14,15]. In addition to maternal malnutrition being a direct determinant of stunting, fulfilling nutrition during pregnancy and preventing anemia in pregnant women are no less critical to breaking the chain of stunting in the future [6]. Indirectly, the quality of childcare, which is closely related to the family’s socio-demographic factors, such as work, mother’s education, and socioeconomic status, has a solid contribution to the birth of babies with stunting [18,19,20,21,22].

Papua is one of the regions in the eastern tip of Indonesia with a low Public Health Development Index (PHDI) score. PHDI is a collection of health indicators developed by the Indonesia Ministry of Health that can be easily and directly measured to describe health problems. This set of health indicators can directly or indirectly play a role in increasing the life expectancy of a long and healthy life [23]. The stunting rate in Papua is higher than the stunting rate in Indonesia. It is recognized that regional disparity is a factor that cannot be omitted when analyzing the stunting rate in Indonesia [24]. Besides, the practice of local culture in Papua also presents a challenge to resolving stunting. Several previous studies found local cultural practices related to child values, family values, taboo foods for infants and children, including forbidden foods for pregnant and lactating women [25,26,27]. Based on the background, the study aimed to analyze the most appropriate maternal characteristics for stunting prevention policies in Papua, Indonesia.

## 2. Materials and Methods

### 2.1. Data Source

The research used secondary data from the 2017 Indonesian Nutritional Status Monitoring Survey. The 2017 Indonesian Nutritional Status Monitoring was a countrywide cross-sectional survey performed by the Indonesian Ministry of Health’s Directorate of Nutrition [28]. The study includes all children under five in two Indonesian provinces (West Papua Province and Papua Province). The study’s unit of analysis was children under the age of five (60 months), and the respondents were mothers. The weighted sample was 11,887 under fives using the multi-stage cluster random sampling method.

### 2.2. Outcome Variable

The study employed stunted children under five years of age as an outcome variable. Stunting was a nutritional status indicator based on height for a child who is reached at a certain age. Based on WHO growth standards, the height indicator for a period is determined based on the z-score or height deviation from average height. Children under five years were separated into two categories: normal and stunted. The limit for the nutritional status category according to the height index/age is [22]:-Stunted: <−3.0 SD to −2.0 SD-Normal: ≥−2.0 SD

### 2.3. Exposure Variables

The research used maternal characteristics as an exposure variable. Maternal characteristics consist of maternal education level, maternal age, maternal marital status, and maternal employment. Additionally, the survey determined maternal education based on the last certificate held by mothers under five. Maternal education consists of the following four levels: primary school and under, junior high school, senior high school, and college. The study determined maternal age based on the last birthday (in years). Furthermore, there are three kinds of maternal marital status, as follows: never married, married, and widowed/divorced. Moreover, maternal employment status consists of two categories: unemployed and employed.

### 2.4. Control Variables

The study analyzed four other control variables, namely, the type of residence, region, age, and gender of children under five. There were two types of residence considered, either urban and rural, and the region consists of two provinces, namely West Papua and Papua. The study determines the age of under-five based on the last month’s birthday (in months). Meanwhile, the gender of children under five years consists of two types, either boy or girl.

### 2.5. Data Analysis

In the early stages of analysis, a co-linearity test was carried out. Then, the Chi-Square test was used to test the dichotomous variables, while the T-test was used for continuous variables. Additionally, a statistical test was conducted to assess whether there is a statistically significant relationship between the variable nutritional status of under five years and the dependent and independent variables. In the final stage, a multivariable test by utilizing a binary logistic regression test was performed.

Moreover, ArcGIS 10.3 (ESRI Inc., Redlands, CA, USA) was used to create a distribution map of stunted under-five by the regency/city in Papua, Indonesia. A shapefile of administrative boundary polygons by the Indonesian Bureau of Statistics was utilized for the task.

### 2.6. Ethical Approval

The 2017 Indonesia Nutritional Status Monitoring Survey has an ethical license approved by the national ethics committee (Number: LB.02.01/2/KE.244/2017). The survey used informed consent during data collection, which accounted for aspects of the procedure for data collection, voluntary and confidentiality.

## 3. Results

The analysis results demonstrate that the proportion of stunted children among Papuan under-fives in Indonesia is 33.1%. Moreover, as shown in Figure 1, no specific pattern based on spatial distribution was found.

Table 1 shows the co-linearity tests’ results, indicating no collinearity between independent variables. Based on Table 1, the analysis results show that the tolerance value for all variables is more significant than 0.10. At the same time, the value of variance inflation factor (VIF) for all variables is less than 10.00. Then, referring to the basis of decision making in the multicollinearity test, no indication of a strong relationship between two or more independent variables in the regression model was found.

### 3.1. Descriptive Analysis

Table 2 provides a statistical description of the under-five characteristics that are the object of analysis in this study. In terms of the type of residence, children under five years who live in rural areas occupy both nutritional status categories. Based on region, children under five living in Papua Province led in both nutritional statuses.

With regard to to maternal education, mothers with primary school education and under led in both nutritional status categories. Based on maternal age, stunted children under five have mothers with an average age slightly older than normal children under five years.

Regarding maternal marital status, married mothers led in both nutritional status categories; on the other hand, according to maternal employment status, unemployed mothers governed in both nutritional status categories.

Table 2 shows that, on average, children under five who are stunted have an older age than children under five who have normal nutritional status. Moreover, based on children under five gender, the girls led in the stunted category; in contrast, there was a greater number of boys in the normal category.

### 3.2. Multivariable Analysis

Table 3 shows the results of the binary regression logistics used to analyze the association between maternal characteristics and stunting among children under five in Papua, Indonesia. The study used the nutritional status “normal” category as a reference in this analysis.

Based on maternal education, mothers who graduated from primary school and under were 1.263 times more likely than mothers with a college education to have stunted under-five children (AOR 1.263; 95% CI 1.228–1.300). Mothers who graduated from junior high school are 1.222 times more likely than mothers with a college education to have stunted children under five years of age (AOR 1.222; 95% CI 1.184–1.262). Moreover, mothers who graduated from senior high school were found to be 1.122 times more likely than mothers with a college education to have stunted under-five children (AOR 1.122; 95% CI 1.089–1.156). This analysis shows that the higher the mother’s education level, the less likely she is to have a stunted under-five children in Papua, Indonesia.

Table 3 shows that maternal age is one of the determinants of stunted under-five children in Papua, Indonesia. Based on maternal marital status, the mother with the never-married group has a 1.138 times higher probability than a divorced/widowed mother of stunted under-five children (AOR 1.138; 95% CI 1.049–1.235). Meanwhile, married mothers are 0.936 times more likely than divorced/widowed mothers to have stunted under five children (AOR 0.936; 95% CI 0.888–0.987). This information shows that maternal marital status determines stunted under-five children in Papua, Indonesia. Married mothers have the lowest probability of having stunted under-five children in Papua, Indonesia.

In addition to the three maternal characteristics, three other control variables also influenced stunted under-five children in Papua, Indonesia. The three characteristics are the type of residence, age of under five, and gender.

Table 3 shows that under five living in urban areas are 0.890 times less likely than under five in rural areas to be stunted (AOR 0.890; 95% CI 0.875–0.905). Based on the under-five gender, the boy has 1.296 times more chance than the girl to be stunted (AOR 1.296; 95% CI 1.278–1.315).

## 4. Discussion

The results show that the higher the maternal education level, the less likely she will have stunted under-five children in Papua, Indonesia. Mothers with a higher level of education have a greater awareness of their children’s health as it is known that mothers with improved schools have nutrition knowledge; this can be beneficial for good feeding behavior [26,29,30]. Mothers with excellent education will use health facilities to meet their children’s food and health needs [31,32]. Better educated mothers know what is best for their children. The study results align with previous research that illustrates that better maternal education reduces the risk of stunting in children aged less than five years [33,34].

The study result found that maternal age is one of the determinants of stunted under-five children in Papua, Indonesia. Getting pregnant at a young age can affect the fetus in the womb; when the fetus requires many nutrients, the mother also needs adequate nutrition for its ongoing growth. In addition, young mothers tend not to access antenatal care services or do not to review their pregnancy with health workers due to their limited knowledge and low levels of education. Therefore, young mothers can give birth to low birth weight babies. Young mothers also have psychologically immature mindsets that impact their child’s upbringing [35,36]. In addition, mothers at young age are at risk of not providing exclusive breastfeeding because of the low supply of breast milk, resulting in stunting in children [37]. The results of this study are in line with previous studies that found a relationship between maternal age and the incidence of stunting [38,39]. However, different results were found in other studies, which found no relationship between maternal age and the incidence of stunting [29].

The results show that maternal marital status determines stunted under-five children in Papua, Indonesia. Marital status is related to the roles of husband and father. Husbands provide emotional support for mothers during the pregnancy and during the child’s growth [40,41]. This emotional support affects the parenting and behavior of mothers in giving food, so that can affect the incidence of stunting in children. In addition, marital status is also related to the father’s role in the child. A father has the same responsibility to nurture, educate, and protect his child. Fathers have a role in perfecting parenting and providing different parenting from mothers. The role of the father and mother will later offer a sense of security, comfort, love, and peace for a child so that it has an impact on the child’s development [42]. Thus, fathers provide care, especially in the first 1000 days of life, by strengthening nutritional needs, implementing a healthy lifestyle, and establishing a quality family life to prevent stunting in children [43]. The study results are in line with previous research that there is a relationship between maternal marital status and the incidence of stunting [22,44]. However, other studies show different results that there is no relationship between maternal marital status and the incidence of stunting [45].

On the other hand, the study found that three control variables also influence the incidence of stunted under-five children in Papua, Indonesia. These variables are type of residence, age of under five, and gender of under five.

Living in urban areas can prevent toddlers from experiencing stunting in Papua, Indonesia. The condition of the urban regions in terms of socioeconomic, diverse food availability, sanitation, education, and access to health facilities is better than in rural areas. Poor sanitation and lack of health programs in rural areas can stunt children [46,47]. In addition, residents tend to have sufficient income to provide a good and decent life for their children in urban areas, primarily through fulfilling their nutritional needs. Education levels and access to health facilities are also better in urban areas. The situation affects the acquisition of health information that can impact feeding behavior in toddlers to affect the incidence of stunting in children [48]. Therefore, increasing people’s standard of living in rural areas by improving the quality of houses, cleanliness, and sanitation can reduce the incidence of stunting in children in rural areas. This study is in line with previous studies that place of residence affects the incidence of stunting [31,49].

The child’s age affects the incidence of stunting in Papua, Indonesia. The condition can be caused by age-related to complementary feeding after six months [31]. In addition, children aged 36–47 months are in a developmental stage with the characteristics of moving actively and putting anything in their mouths. The situation can increase the risk of other infections to stunting [50]. The results of this study are in line with previous studies that children’s age affects the incidence of stunting [29,33,51]. The gender of the child affects the incidence of stunting in Papua, Indonesia. Boys are more prone to stunting than girls. This is because boys require more calories for growth and development [31]. If these calories cannot be appropriately met, growth will be stunted to cause stunting. The study results align with previous studies that there is a relationship between gender and the incidence of stunting [29,33,51].

### Study Limitation

The authors analyzed the 2017 Indonesia Nutritional Status Monitoring Survey using secondary data. The variables analyzed were limited to those provided by the survey. The analytical results cannot explain several other variables that have been identified from previous studies to affect stunting in children under five, including antenatal care, maternal stature, maternal body mass index, wealth index, diarrhoea, anaemia, and agri-food [52,53,54,55].

On the other hand, the study conducted with a quantitative approach, which cannot explain the influence of cultural factors that are still very strong in Indonesia, especially in rural areas. Several previous studies reported results related to these cultural factors, including the value of children, food taboo, parenting, and intake patterns [27,30,56].

## 5. Conclusions

Based on the research results, the study concludes that maternal education, maternal age, and maternal marital status are associated with stunting in children under-five in Papuan in Indonesia. The target for stunting prevention policies is mothers with poor education and who are single.

Further studies using a qualitative approach are needed to explore the phenomena found about the mother’s characteristics in terms of poor education and being single in Papua, commonly used in child care. Furthermore, values and views of the Papuan people of mothers with poor education who are single are of interest.

## Figures and Tables

**Figure 1 nutrients-14-00549-f001:**
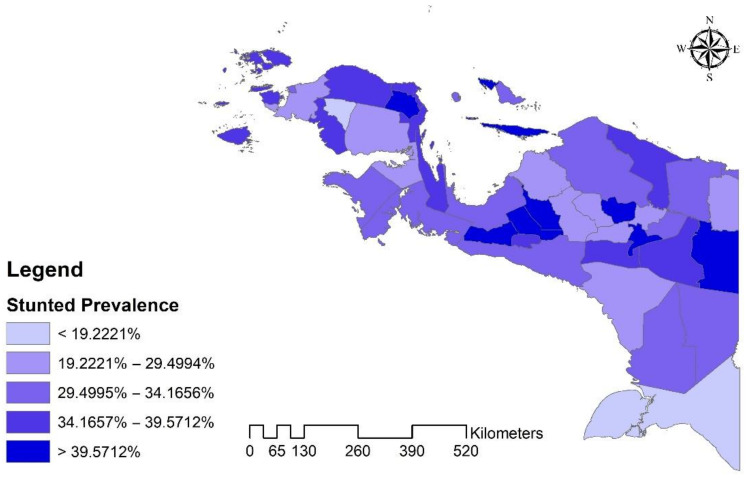
Distribution Map of Under Five Stunted by the Regency/City in Papua, Indonesia.

**Table 1 nutrients-14-00549-t001:** The results for the co-linearity test of Nutritional Status among Papuan Under Five in Indonesia.

Variables	Collinearity Statistics	
	Tolerance	VIF
**Area context**		
Residence	0.843	1.186
Region	0.919	1.088
**Maternal Characteristics**		
Mother’s Education level	0.865	1.156
Mother’s Age (in years)	0.959	1.043
Mother’s Marital status	0.991	1.009
Mother’s Employment Status	0.907	1.102
**Under fives’ Characteristics**		
Age	0.969	1.032
Gender	0.999	1.001

Dependent Variable: Nutritional status of the toddler.

**Table 2 nutrients-14-00549-t002:** Descriptive Statistic of Nutritional Status among Papuan Under-five in Indonesia.

Variables	Nutritional Status		*p*-Value
	Normal (*n* = 7840)	Stunted (*n* = 4047)	
**Residence**			* 0.000
• Urban	30.9%	35.1%	
• Rural	69.1%	64.9%	
**Region**			0.052
• West Papua	24.8%	24.9%	
• Papua	75.2%	75.1%	
**Maternal Characteristics**			
• Education level			* 0.000
• Primary school and Under	49.3%	45.2%	
• Junior high school	17.0%	17.0%	
• Senior high school	26.8%	29.6%	
• College	6.9%	8.3%	
Age (in years; mean)	(28.51)	(28.63)	* 0.000
Marital status			* 0.000
• Never married	1.4%	1.1%	
• Married	96.8%	97.2%	
• Divorce/Widowed	1.9%	1.7%	
Employment Status			* 0.000
• Unemployed	56.6%	57.9%	
• Employed	43.4%	42.1%	
**Under fives’ Characteristics**			
Age (in months; mean)	(24.83)	(31.26)	* 0.000
Gender			* 0.000
• Boy	53.9%	47.7%	
• Girl	46.1%	52.3%	

Note: ∗ *p* < 0.001.

**Table 3 nutrients-14-00549-t003:** Binary Logistic Regression of Nutritional Status among Papuan Under Five in Indonesia.

Predictors	Stunted			
	*p*-Value	AOR	95% CI	
			Lower Bound	Upper Bound
Residence: Urban	*** 0.000	0.890	0.875	0.905
Residence: Rural	-	-	-	-
Maternal Education: Primary school and under	*** 0.000	1.263	1.228	1.300
Maternal Education: Junior high school	*** 0.000	1.222	1.184	1.262
Maternal Education: Senior high school	*** 0.000	1.122	1.089	1.156
Maternal Education: College	-	-	-	-
Maternal age	*** 0.000	0.994	0.993	0.995
Maternal Marital Status: Never married	** 0.002	1.138	1.049	1.235
Maternal Marital Status: Married	* 0.014	0.936	0.888	0.987
Maternal Marital Status: Divorced/widowed	-	-	-	-
Maternal employment: Unemployed	0.804	1.002	0.987	1.017
Maternal employment: Employed	-	-	-	-
Under fives’ age	*** 0.000	1.023	1.023	1.024
Under fives’ Gender: Boy	*** 0.000	1.296	1.278	1.315
Under fives’ Gender: Girl	-	-	-	-

Note: AOR: Adjusted Odds Ratio; CI: Confidence Interval; * *p* < 0.050; ** *p* < 0.010; *** *p* < 0.001.

## Data Availability

The 2017 Nutrition Status Monitoring Survey data used to support these findings of this study were supplied by the Directorate of Community Nutrition of the Indonesian Ministry of Health under license and so cannot be made freely available. Requests for access to these data should be made to the Directorate of Community Nutrition of the Indonesian Ministry of Health.

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
