# Peer review of "The Targets for Stunting Prevention Policies in Papua, Indonesia: What Mothers’ Characteristics Matter?"

_nutrients, 2022, doi:10.3390/nu14030549_

Round 1

Reviewer 1 Report

This manuscript addresses an extremely important topic about stunting in children, which interferes with a proper growth and development, and has a negative impact in the children’s future and productivity as adults. In order to start preventing these issues, the authors analysed the appropriate maternal characteristics for stunting prevention policies in the region of Papua, Indonesia.

The manuscript could be considered for publication, but authors should be prepared to incorporate some major revisions. Also, English language must be improved (for instance, line 45 “It is, of course, will contribute to the burden of diseases due to stunting.”, and line 69 “In addition to maternal malnutrition during pregnancy, it is also a direct determinant of stunting”).

Introduction provides a good explanation of the problem and globally is clear, however in line 75 the IPKM score has to be explained. What is the IPKM score?

Materials and Methods seems to me clear, but English language has to be improved, for instance the sentence in line 111 “ apart from…”.

Results section also need to be improved for a clear understanding. The first sentence is very confused, as well as the following paragraph. Section 3.1 is also very confused and it must provided an adequate extended description of the sample characteristics. I understand that stunted children are in higher proportion in the urban region than in the rural region, but where do you write about that?
You have to be prepared also to make a good discussion about this result. Regarding the education level, the author only commented the primary school level, however the proportion of stunted children are higher in mothers with senior level and college. These results are as important as the primary level education, as they represent almost the other half of the studied sample.

I found the 'Discussion' section to lack some important issues about the results, with clarity and clean language. In my perspective, it is not enough to compare the results of this work, with the other published studies just stating, “The results of this study are in line…” and “other studies show different results…”. There is a need for a more profound analyzes to support and provide a strong discussion. My main recommendation is to rephrase almost all the ‘Discussion’ section.

Author Response

Reviewer 1

  1. This manuscript addresses an extremely important topic about stunting in children, which interferes with a proper growth and development, and has a negative impact in the children’s future and productivity as adults. In order to start preventing these issues, the authors analysed the appropriate maternal characteristics for stunting prevention policies in the region of Papua, Indonesia.
  2. The manuscript could be considered for publication, but authors should be prepared to incorporate some major revisions. Also, English language must be improved (for instance, line 45 “It is, of course, will contribute to the burden of diseases due to stunting.”, and line 69 “In addition to maternal malnutrition during pregnancy, it is also a direct determinant of stunting”).

The revised manuscript was through a proofread process.

  1. Introduction provides a good explanation of the problem and globally is clear, however in line 75 the IPKM score has to be explained. What is the IPKM score?

IPKM or PHDI (Public Health Development Index) is a collection of health indicators developed by the Indonesia MOH that can be easily and directly measured to describe health problems. This set of health indicators can directly or indirectly play a role in increasing the life expectancy of a long and healthy life.

  1. Materials and Methods seems to me clear, but English language has to be improved, for instance the sentence in line 111 “ apart from…”.

The revised manuscript was through a proofread process.

The study analyzed four other control variables: the type of residence, region, age, and gender of under five.

  1. Results section also need to be improved for a clear understanding. The first sentence is very confused, as well as the following paragraph. Section 3.1 is also very confused and it must provided an adequate extended description of the sample characteristics. I understand that stunted children are in higher proportion in the urban region than in the rural region, but where do you write about that?

Table 1 shows the stunted children in higher proportion in the rural region.You have to be prepared also to make a good discussion about this result. Regarding the education level, the author only commented the primary school level, however the proportion of stunted children are higher in mothers with senior level and college. These results are as important as the primary level education, as they represent almost the other half of the studied sample.

The study compared among education level more detailed in results about Table 3 (page 5-6).

  1. I found the 'Discussion' section to lack some important issues about the results, with clarity and clean language. In my perspective, it is not enough to compare the results of this work, with the other published studies just stating, “The results of this study are in line…” and “other studies show different results…”. There is a need for a more profound analyzes to support and provide a strong discussion. My main recommendation is to rephrase almost all the ‘Discussion’ section

The discussion was revised as suggested.

Reviewer 2 Report

  1. The introduction part do not match the aim and the result of this paper, for the author want to get maternal characteristics for children stunting prevention policies. you beed give the evidence that education, family income, the current  mother marital status or single related may influence the babiies stunting outcome;
  2. Although this paper used a finished survey data,  a secondary analysis, I suggest you must give a new method description, in this manuscript, just like variables and outcome data analysis.
  3. the results forms ad description are not line with other paper ublished at nutrients;
  4. discussion: Focus on the data analysis more than to explain the results and compare to other studies, what's new or what's the reason for it.
  5. Please revised the paper totally, for the english native expression.

Author Response

Reviewer 2

  1. The introduction part do not match the aim and the result of this paper, for the author want to get maternal characteristics for children stunting prevention policies. you beed give the evidence that education, family income, the current  mother marital status or single related may influence the babiies stunting outcome;

The evidence is already in the narrative introduction.

Indirectly, the quality of child care, which is closely related to the family's socio-demographic factors, such as work, mother's education, and socioeconomic status, has a solid contribution to the birth of babies with stunting.

  1. Although this paper used a finished survey data,  a secondary analysis, I suggest you must give a new method description, in this manuscript, just like variables and outcome data analysis.

The methods was revised as suggested.

  1. the results forms ad description are not line with other paper ublished at nutrients;

The authors feel that they have written the manuscript according to the Nutrients format.

  1. discussion: Focus on the data analysis more than to explain the results and compare to other studies, what's new or what's the reason for it.

The discussion was revised as suggested.

  1. Please revised the paper totally, for the english native expression.

The revised manuscript was through a proofread process.

Round 2

Reviewer 1 Report

Although the manuscript has improved, in particular the clarity of the results and discussion section, English language needs more. Thus, I suggest the authors to read carefully the manuscript and improve it, and also to make the sentences "stick" together better.

Be careful with the use of the sentence connectors: to have coehsion over the manuscript – for instance – line 190 “meanwhile”.

Lines 231-233 “Mothers with better education have a higher awareness of their children's health as it is known that mothers with better education have nutrition knowledge”, and again in lines 234-236 “Mothers with better education will also use health facilities to obtain their children's nutritional and health needs [31,32]. Mothers with better education know what the best for her child. “

It’s too repetitive … “better education” . I suggest to rephrase the paragraph.

Line 241 - "ANC services"… please explain it

Lines 241- 242 “due to their limited knowledge due to low levels of education. “I suggest to rephrase the sentence: due… due repetition.

Line 285 – please use “gender” instead of “sex”

 Lines 302-303- “Based on the research results, the study concluded maternal education, maternal age, and maternal marital status associated with stunted under-five Papuan in Indonesia” - lacks the correct verb form “are associated”

Conclusion section is weak.
What are your future research intentions in order to make changes or to fulfil the lacks you found in the current scenario? And how do you think you can do it? Do you think it is important a new research or collect new data to achieve a better scenario? Have you suggestions for future studies that need to be carried out?

Author Response

Reviewer 1

Although the manuscript has improved, in particular the clarity of the results and discussion section, English language needs more. Thus, I suggest the authors to read carefully the manuscript and improve it, and also to make the sentences "stick" together better.

Be careful with the use of the sentence connectors: to have coehsion over the manuscript – for instance – line 190 “meanwhile”.

The author was revised the sentence connector to “on the other hand”.

Regarding maternal marital status, married mothers led both nutritional status categories; on the other hand, according to maternal employment status, unemployed mothers governed in both nutritional status categories.

Lines 231-233 “Mothers with better education have a higher awareness of their children's health as it is known that mothers with better education have nutrition knowledge”, and again in lines 234-236 “Mothers with better education will also use health facilities to obtain their children's nutritional and health needs [31,32]. Mothers with better education know what the best for her child. “

It’s too repetitive … “better education” . I suggest to rephrase the paragraph.

The author rephrased the paragraft as suggested.

Mothers with better education have a higher awareness of their children's health as it is known that mothers with improved schools have nutrition knowledge; this can be beneficial for good feeding behavior [26,29,30]. Mothers with excellent education will use health facilities to meet their children's food and health needs [31,32]. Better educated mothers know what is best for their children.

Line 241 - "ANC services"… please explain it

The author added narration in that term.

In addition, young mothers tend not to access antenatal care services or check their pregnancy to the health workers due to their limited knowledge and low levels of education.

Lines 241- 242 “due to their limited knowledge due to low levels of education. “I suggest to rephrase the sentence: due… due repetition.

The manuscript was revised as suggested.

In addition, young mothers tend not to access antenatal care services or check their pregnancy to the health workers due to their limited knowledge and low levels of education.

Line 285 – please use “gender” instead of “sex”

The manuscript was revised as suggested.

 Lines 302-303- “Based on the research results, the study concluded maternal education, maternal age, and maternal marital status associated with stunted under-five Papuan in Indonesia” - lacks the correct verb form “are associated”

The manuscript was revised as suggested.

Conclusion section is weak.
What are your future research intentions in order to make changes or to fulfil the lacks you found in the current scenario? And how do you think you can do it? Do you think it is important a new research or collect new data to achieve a better scenario? Have you suggestions for future studies that need to be carried out?

The manuscript was revised as suggested.

Further studies with a qualitative approach are needed to explore the phenomena found: about the mother's behavior with poor education and single in Papua, which her commonly used in child care; about the values and views of the Papuan people on the mother with poor education and single.
